# High-Dose Spermidine Supplementation Does Not Increase Spermidine Levels in Blood Plasma and Saliva of Healthy Adults: A Randomized Placebo-Controlled Pharmacokinetic and Metabolomic Study

**DOI:** 10.3390/nu15081852

**Published:** 2023-04-12

**Authors:** Stefan Senekowitsch, Eliza Wietkamp, Michael Grimm, Franziska Schmelter, Philipp Schick, Anna Kordowski, Christian Sina, Hans Otzen, Werner Weitschies, Martin Smollich

**Affiliations:** 1Department of Biopharmaceutics and Pharmaceutical Technology, University of Greifswald, 17489 Greifswald, Germany; 2Institute of Nutritional Medicine, University Hospital Schleswig-Holstein, Campus Lübeck, University of Lübeck, 23538 Lübeck, Germany

**Keywords:** spermidine, spermine, putrescine, autophagy, SARS-CoV-2, COVID-19

## Abstract

(1) Background: Spermidine is a biogenic polyamine that plays a crucial role in mammalian metabolism. As spermidine levels decline with age, spermidine supplementation is suggested to prevent or delay age-related diseases. However, valid pharmacokinetic data regarding spermidine remains lacking. Therefore, for the first time, the present study investigated the pharmacokinetics of oral spermidine supplementation. (2) Methods: This study was designed as a randomized, placebo-controlled, triple-blinded, two-armed crossover trial with two 5-day intervention phases separated by a washout phase of 9 days. In 12 healthy volunteers, 15 mg/d of spermidine was administered orally, and blood and saliva samples were taken. Spermidine, spermine, and putrescine were quantified by liquid chromatography–mass spectrometry (LC–MS/MS). The plasma metabolome was investigated using nuclear magnetic resonance (NMR) metabolomics. (3) Results: Compared with a placebo, spermidine supplementation significantly increased spermine levels in the plasma, but it did not affect spermidine or putrescine levels. No effect on salivary polyamine concentrations was observed. (4) Conclusions: This study’s results suggest that dietary spermidine is presystemically converted into spermine, which then enters systemic circulation. Presumably, the in vitro and clinical effects of spermidine are at least in part attributable to its metabolite, spermine. It is rather unlikely that spermidine supplements with doses <15 mg/d exert any short-term effects.

## 1. Introduction

Spermidine is a biogenic polyamine that plays a crucial role in cellular homeostasis, cell growth, proliferation, and autophagy [1,2,3]. Research has found a relevant role for biogenic polyamines in both health and disease [4,5]. The physiological spermidine pool is fed by cellular de novo synthesis, intestinal microbiota biosynthesis with subsequent absorption, and dietary sources [6]. In mammalian cells, spermidine is generated from its precursor putrescine or through the degradation of spermine [7], and different mechanisms of transmembrane transport contribute to the regulation of intracellular spermidine concentrations [8].

Polyamines (including spermidine) in the gastrointestinal lumen originate from food, intestinal microbiota, pancreatic-biliary secretions, and degraded intestinal cells [9,10], and dietary polyamines are proposed to be the main source of luminal polyamines [8].

The average daily nutritional intake of spermidine in European countries varies from approximately 10 to 15 mg [11]. Examples of foods with very high spermidine concentrations are cheddar cheese (200 mg/kg) and mushrooms (90 mg/kg) [12]. The dietary sources of individual spermidine intake vary widely due to individual food preferences and regional dietary habits [13,14]. The main dietary sources of spermidine are potatoes (6.4% of total), vegetable sprouts (7.3%), salad (9.8%), pears and apples (13.3%), and whole grains (13.4%) [13].

To date, little is known about the mechanisms of intestinal absorption, presystemic metabolism, and the kinetics of polyamine transport systems in humans [8]. The intestinal absorption of polyamines is complex due to different polyamine transporters [7]. After polyamine-containing foods are ingested, the majority of luminal polyamines disappear from the intestinal lumen through either passive diffusion or intraluminal degradation [15,16]. Although dietary polyamines are found in millimolar concentrations in the duodenum and jejunum after a high-dose intake, the plasma concentration reaches a maximum of 10–20 μM. Furthermore, 120 min after a polyamine-rich meal, polyamine concentrations in the intestinal lumen return to preprandial baseline levels. Thus, only low concentrations are reached in the blood (μM) despite high concentrations in food (mM) [17]. In rodent models, after oral intake, radioactively labeled dietary polyamines have been reported to reach all organs, the skeletal muscles, and even tumor tissues [18,19].

Luminal polyamines are (at least in part) actively imported into intestinal epithelial cells (IECs) through the polyamine transport system (PTS) [20]. In IECs, the polyamines may be converted by polyamine metabolic enzymes or emitted across the basolateral membrane to the systemic circulation [15]. Moreover, paracellular passive absorption may also contribute to the uptake of luminal polyamines. Additionally, antizymes and antizyme inhibitors regulate cellular polyamine uptake [8]. However, the role of these proteins in intestinal polyamine absorption remains rather speculative [21].

Spermidine has an autophagy-inducing effect that makes it highly interesting for preclinical research [5,22]. As mechanistic studies have demonstrated, spermidine induces autophagy and mitophagy through various mechanisms. The predominant mechanisms are the inhibition of the lysine acetyltransferase EP300 and the induction of a reactive oxygen species (ROS) burst, which leads to the activation of the Ataxia-telangiectasia mutated (ATM) protein [6].

Most prominent translational research with spermidine has focused on its possible efficacy in the context of brain health/aging, oncology, and SARS-CoV-2 infections. 

Regarding brain health and aging, spermidine-induced autophagy has been postulated to exert protective effects on neurodegenerative disorders and subjective cognitive decline and also to slow aging processes even in humans [23,24,25]. However, the role of spermidine in brain aging and age-related cognitive decline is complex. On the one hand, supplementation with spermidine may support healthy aging, while on the other hand, elevated spermidine plasma levels are associated with advanced brain aging, and they might serve as a potential early biomarker for Alzheimer’s disease and vascular brain pathology [26].

In fact, the administration of exogenous spermidine extends the lifespan in various models of aging through epigenetic modifications, autophagy induction, and necrosis suppression [27]. In animal models, spermidine supplementation has had beneficial effects on brain and cognitive health [5]. However, clinical data failed to confirm any positive effects of spermidine on age-related cognitive decline in contrast to preclinical studies, implying a respective effect [28]. The same also applies to the putative role of spermidine in other age-related diseases [8,29], cardiovascular health [30,31,32], nonalcoholic steatohepatitis (NASH) [33], Alzheimer’s disease [34], immune functions [35], and ophthalmology [36].

In oncology, due to its direct link with oncogenes, polyamine metabolism has long been a target for potential cancer therapeutic agents [37]. For example, recent data indicate that polyamines could play a major role in regulating the antitumor immune response [7]. Furthermore, in vivo data suggest that the efficacy of antineoplastic chemotherapies might be increased by 48-h fasting periods prior to drug application [23]. The hypothetical mechanism is fasting-mediated autophagy induction. If respective effects can be confirmed in clinical trials, autophagy-inducing compounds such as spermidine might be suitable for use in patients with contraindications against fasting. To date, however, the role of autophagy in cancer remains controversial [38]. 

Recently, in vitro data revealed that an infection with SARS-CoV-2 leads to the viral inhibition of spermidine synthase; as a result, the intracellular spermidine concentration decreases, and autophagy as a cellular defense mechanism is reduced [38]. Incubation of SARS-CoV-2-infected Vero cells with 333 µM spermidine reduced viral replication by 85%; furthermore, spermidine preincubation prior to SARS-CoV-2 infection reduced viral replication by 70% [38]. Based on these data, it has been hypothesized that spermidine supplementation might have a clinically relevant antiviral effect on a SARS-CoV-2 infection [39]. This hypothesis was endorsed by more current experimental results, which suggested that spermidine might play a central role in COVID-19 pathogenesis [40]. 

To probe these preclinical data in future clinical trials, detailed knowledge of spermidine pharmacokinetics is essential. Since SARS-CoV-2 replicates predominantly in the pharynx during the initial phase of infection, saliva concentrations of spermidine (as well as spermine and putrescine) after oral intake of spermidine are of major interest. Therefore, in addition to the blood plasma concentrations, we also analyzed the respective saliva concentrations. To the best of our knowledge, the concentrations of spermidine, spermine, and putrescine in saliva after oral intake of spermidine have never been investigated before.

Thus, to foster translational research with spermidine, we determined the pharmacokinetics of orally administered spermidine, its precursor putrescine, and its metabolite spermine, as well as the effect of its multiple-day administration on the steady-state concentrations in blood and saliva in healthy volunteers.

Additionally, also for the first time, we investigated the effect of orally administered spermidine on the human plasma metabolome using nuclear magnetic resonance (NMR) spectroscopy because spermidine has been postulated to act as a calorie restriction mimetic (CRM) in humans, which would imply effects on the plasma concentrations of various metabolites [41,42].

## 2. Materials and Methods

### 2.1. Study Design

The study was designed as a randomized, placebo-controlled, triple-blinded, two-armed crossover trial with a washout phase of 9 days between the two intervention phases (Figure 1). The first intervention phase took place from Monday to Friday, followed by the washout phase (the weekend and following week), and then the second intervention phase started on the following Monday.

The subjects were recruited through the University of Luebeck’s email distribution list. Both supplementation and blood/saliva sampling were performed at the Institute of Nutritional Medicine. Randomization, blinding, and the production of the placebo and verum preparations were provided by the Center of Drug Absorption and Transport (C_DAT) of the University of Greifswald, Germany. Randomization was performed using random numbers generated in Microsoft Excel. With the command = random number(), a random number between 0 and 1 was generated for each subject. Subsequently, a comparison was made in blocks of two to determine which subject had the higher random number. Those with the higher random number started with the verum arm, while subjects with the lower random number started with the placebo arm. This procedure ensured an equal distribution (balance) of the two possible orders (V-P or P-V) among the 12 subjects. Thus, 6 subjects each start with verum or placebo. The capsules taken (verum or placebo) were provided by staff in Greifswald on the basis of the generated sequence for weeks 1 and 2, specifically for the subject-specific random number. Externally, verum and placebo are indistinguishable due to color-matched filling powder and capsules of the same size and color. For more information about products and capsules, see Section 2.3. The containers were assigned on the basis of pseudonym and subject number as well as treatment week. Thus, at the study site in Lübeck, the performing staff as well as the subjects were blinded to the assignment and only received the respective prepared capsule container for week 1 or week 2. Until the analyses were completed, the samples and results were only assigned to week 1 or week 2 of the subjects. Only after the completion of the measurements was the unblinding carried out using the randomization list. Thus, analytical staff, clinical personnel, and subjects were blinded to the randomization order.

To detect possible dietary changes in the course of the study that might have influenced the results, the participants documented their meals throughout the study period in a validated nutritional log. Throughout the study course, subjects were advised to refrain from extraordinary physical work, dietary changes, and alcohol consumption of more than two standard glasses per day. In addition, the participants were advised to follow their usual activity and dietary habits.

As this study was designed as an exploratory pharmacokinetic trial, case number calculation was not possible due to a lack of data. A sample size of 12 was considered appropriate as it is established in phase 1 clinical trials. Moreover, the robustness of the data was improved by the cross-over design with placebo control.

The study was registered in the German Clinical Trials Register (DRKS) (DRKS00029397).

### 2.2. Study Cohort and Ethics

Study participants were recruited between December 2021 and April 2022. The inclusion criteria were age 18–45 and sufficient knowledge of the German language to understand the study instructions. The exclusion criteria were BMI < 18 or >29.9 kg/m^2^, acute diseases, a history of food allergy, intake of any medication, and dietary supplements during the 48-h period before the start of the study until the end of the study (except oral contraceptives), use of illegal substances, pregnancy and breast-feeding, shift work, and adherence to a vegan diet.

In total, 85 subjects were assessed for eligibility (Figure 2). After applying the inclusion and exclusion criteria and clarifying the study details, 12 subjects (eight females, four males) agreed to participate and were randomized accordingly. Informed consent was obtained from all subjects. All participants completed the study schedule. The median age of the female participants was 24 years (21–45 years), while that of the male participants was 26 years (22–28 years).

### 2.3. Spermidine and Placebo Preparation

Commercially available dietary supplements of the brand MoleQlar™ (Rottach-Egern, Germany) based on *Chlorella algae* powder and soybean extract were purchased, each containing 1.5 mg of spermidine per capsule. The capsules were vegan and consisted of hydroxypropyl methylcellulose (HPMC). The content indicated on the label was verified by an independent laboratory. Five administered hard capsules corresponded to a single dose of 7.5 mg of spermidine. Similar HPMC capsules (Vcaps Plus, Lonza Capsules & Health Ingredients, Bornem, Belgium) were used for the placebo and filled with inert and similar colored capsule filler consisting of microcrystalline cellulose (MCC Pharmacell^®^ 102, DFE Pharma, Goch, Germany) and iron oxide (Ferrum oxydatum flavum, Caesar & Loretz, Hilden, Germany).

### 2.4. Intervention

The intervention started with a standard dinner the evening before the first measurement day (day 0 and day 4; Figure 3).

The next morning (measurement day; day 1 and day 5), placebo or verum capsules were administered to the subjects. The target dose in the verum arm of the study was 15 mg of spermidine per day (corresponding to five capsules in the morning and five in the evening). The same number of capsules were administered in the placebo arm. Subsequently, during the following 8 h, blood samples (9 mL of whole blood per sample) and saliva samples (after each blood sample) were taken under standardized dietary conditions. For the next 3 days, the respective capsules (verum or placebo) were self-administered by the participants every morning and evening, with one preprandial sampling in the morning. The second measurement day was on day 5 (similar to day 1). After a wash-out phase of 9 days, the second intervention phase started similarly to the first intervention phase.

The water intake during the measurement days was standardized until noon. The participants drank 100 mL of non-carbonated water every full hour, directly after blood and saliva samples were taken. After lunch, they had water ad libitum.

The calculation of pharmacokinetics was based on all blood and saliva samples. By contrast, only six samples of blood per subject were analyzed for the NMR metabolomics measurements (day 1 before breakfast, day 1 after snack, and day 5 after snack in both intervention phases; corresponding to samples 1, 11, and 25 or 0 h, 8 h, and 104 h, respectively).

### 2.5. Standard Meals

The use of standard meals ensured that the study participants consumed a comparable amount of polyamines with their food before and during the sampling on the measurement days. The standard meals were taken at the Institute for Nutritional Medicine and were composed as follows:Standard dinner (days 0, 4, 14, and 18): a medium-sized margherita pizza from a local delivery service; non-carbonated water ad libitum.Standard breakfast (days 1, 5, 15, and 19): mixed bread, butter, jam, honey, and orange juice; non-carbonated water ad libitum.Standard lunch (days 1, 5, 15, and 19): a medium-sized salad (lettuce, peppers, cucumber, tomatoes, and yogurt dressing) and one bun; non-carbonated water ad libitum.Standard snack (days 1, 5, 15, and 19): a piece of crumble cake; non-carbonated water ad libitum.

The meals between the measurement days were not standardized. However, study participants were instructed to refrain from relevant dietary changes during the study course. Compliance with this instruction was ensured by reviewing the nutritional logs provided by the participants.

### 2.6. Blood and Saliva Sampling

To ensure the validity of the time-dependent concentration determinations, blood and saliva samples were taken with a maximum time deviation of 5% of the planned time intervals. On days 1, 5, 15, and 19, blood samples were taken using a peripheral venous indwelling catheter placed in the crook of the subject’s elbow. At the end of each study day, the catheter was removed. On the shorter measurement days (days 2, 3, 4, 16, 17, and 18), blood samples were taken using a disposable cannula (butterfly). At any time point scheduled, 9 mL of peripheral blood was taken in an EDTA K3 Monovette^®^ (Sarstedt AG & Co. KG, Nuembrecht, Germany), corresponding to a total of 50 blood samples per subject by the end of the study.

After collection, blood samples were centrifuged at 4 °C and 3000× *g* for 10 min. It was ensured that the samples were never at room temperature for more than 10 min prior to refrigerated centrifugation. Immediately after centrifugation, the plasma supernatant was pipetted into ice-cooled microtubes. For each sample, 1000 μL of plasma was transferred into three different microtubes and frozen at −80 °C. Two samples were used for the time-dependent concentration determination and NMR metabolomics, respectively, and one sample was used as a retained sample.

The saliva samples were placed in appropriate microtubes by the study participants immediately after the respective blood sampling and frozen at −80 °C.

### 2.7. Pharmacokinetics

Putrescine(1,4-diaminobutan) dihydrochloride and spermidine trihydrochloride were obtained from Acros Organics (Geel, Belgium), and spermine tetrahydrochloride was purchased from Alfa Aesar (Haverhill, MA, USA). Putrescine-d4(1,4-diaminobutan-2,2,3,3-d4) dihydrochloride, spermidine-(butyl-d8) trihydrochloride, *N*-(9-Fluorenylmethoxycarbonyloxy)succinimide (Fmoc-OSu), boric acid, and formic acid were obtained from Sigma-Aldrich (St. Louis, MO, USA). Acetonitrile (super gradient grade) was obtained from VWR (Radnor, PA, USA), while sodium hydroxide was provided by AppliChem (Darmstadt, Germany). All solvents (acetonitrile, water) and additives (formic acid) used for LC–MS/MS analysis were of LC–MS grade and obtained from VWR (Radnor, USA).

The determination of the concentrations of spermidine, spermine, and putrescine in plasma and saliva was performed by protein precipitation following derivatization with Fmoc-OSu based on the method of Xiong and Zhai [44]. Isotopically labeled spermidine (spermidine-d8 for spermidine and spermine) and putrescine (putrescine-d4 for putrescine) were used as internal standards.

Stock solutions, quality control spiking solutions, and internal standard solutions were prepared in methanol-water (20:80, *v/v*) and diluted with the same solvent unless stated otherwise. The quantitation range was 2–500 ng/mL for all polyamines in plasma. For saliva, a quantitation range of 10–2500 ng/mL was chosen for spermidine and spermine, while it was 0.4–100 µg/mL for putrescine. Each concentration of putrescine-d4 and spermidine-d8 in the internal standard solution was 2.5 µg/mL for plasma and 100 µg/mL and 2.5 µg/mL, respectively, for saliva analysis.

A detailed overview of the sample preparation can be found in Appendix A. Briefly, the frozen (−80 °C) samples were thawed at room temperature and vortexed (IKA^®^ VORTEX 2, IKA^®^-Werke GmbH & Co. KG, Staufen, Germany). The saliva samples were centrifuged for 10 min at 16,060× *g* (Biofuge^®^ pico, Heraeus, Hanau, Germany), while the plasma samples were directly processed further. An aliquot of 100 µL of plasma or supernatant of centrifuged saliva, respectively, was transferred into a 1.5 mL microcentrifuge tube (Thermo Fisher Scientific, Waltham, MA, USA) and spiked with 25 µL of internal standard solution. Proteins were precipitated by adding acetonitrile containing 0.2% formic acid. After centrifugation for 10 min at 16,060× *g* and the transfer of an aliquot of the supernatant into a 1.5 mL amber glass vial (VWR, Radnor, PA, USA), 200 mM borate buffer with a pH of 9.0 and 5 mM Fmoc-OSu solution in acetonitrile were added. After 30 min of incubation at room temperature, the derivatization reaction was stopped by adding formic acid (100%). This extract was analyzed using a LC–MS/MS system (Shimadzu Corporation, Kyoto, Japan). The LC–MS/MS system consisted of an LC–40B X3 solvent delivery module, an SIL-40C X3 auto sampler, a CTO-40S column oven, an SPD-40 UV-Vis detector, an FCV-20AH_2_ valve unit, and an LC–MS-8060 mass spectrometer equipped with an ESI ionization unit. 

The LC system was equipped with a Phenomenex Kinetex^®^ 2.6 µm PS C18 100 Å 150 × 2.1 mm column (Phenomenex, Torrance, CA, USA) protected by a SecurityGuard^TM^ ULTRA Cartridge (Phenomenex, Torrance, CA, USA), which was connected to a SecurityGuard^TM^ ULTRA Holder (Phenomenex, Torrance, CA, USA). Acetonitrile + 0.1% HCOOH (A) and water + 0.1% HCOOH (B) were used as eluents for the mobile phase at a flow rate of 0.4 mL/min. A linear gradient was applied for the separation of the derivatized polyamines (Appendix A). The oven temperature was set to 40 °C. The total run time of the method was 12 min.

The retention times of spermidine, spermine, and putrescine were 5.60, 6.90, and 3.65 min, respectively. To prevent contamination of the ionization unit, the eluent was initially directed to a waste bottle. For the analysis of plasma samples, after 3.4 min, the valve unit changed the direction of the flow to the mass spectrometer, and after 7.4 min, the valve switched back to its initial position. For the determination of putrescine in saliva, the valve switched the flow to the MS after 3.4 min and back after 4.6 min, while for the measurement of spermidine and spermine in saliva, the eluent was directed to the MS from 5.1 to 7.4 min.

The detection of analytes was performed in the positive multiple reaction monitoring (MRM) mode (Appendix A). Furthermore, the following parameters were applied: nebulizing gas flow 3 L/min, heating gas flow 10 L/min, drying gas flow 3 L/min, interface temperature 300 °C, desolvation temperature 526 °C, desolvation line temperature 250 °C, and heat block temperature 400 °C.

Data acquisition and analysis were performed using LabSolutions (Version 5.97 SP1). The calibration curves consisted of eight calibration standards and were constructed by plotting peak area ratios of the analytes and internal standards against the concentration of the analytes. A 1/c^2^ weighted least squares linear regression was used for all polyamines.

The method was validated with respect to intra- and inter-day accuracy and precision at three concentration levels (QC-L, QC-M, and QC-H) in six replicates on the same day and 18 replicates over 2 days, respectively. Stability (freeze-thaw stability, long-term stability [6 weeks at −80 °C], and short-term stability [3 h at room temperature]) was evaluated at low (QC-L) and high (QC-H) quality control concentrations in four replicates. Furthermore, reinjection stability was investigated at three concentration levels (QC-L, QC-M, and QC-H) in six replicates after storage for 24 h in the autosampler. Plasma and saliva quality control standards were prepared by spiking QC working solutions (comprising 5% of the total volume) into the corresponding matrix. As no analyte-free plasma or saliva were available, the parameters of validation were evaluated by subtracting the concentration of nonspiked samples from spiked matrix samples. The matrix effect was evaluated by spiking the plasma and saliva of six individuals at low (QC-L) and high (QC-H) quality control concentrations and calculating the coefficient of variation (CV) of the measured concentration. As presented in Appendix A, the method proved to be accurate and precise with respect to the aforementioned validation parameters. 

Moreover, the stability investigation (data not presented) proved that the unprocessed plasma samples were stable for two freeze-thaw cycles, while unprocessed saliva samples were stable for three cycles. Unprocessed plasma and saliva samples were stable for a period of 6 weeks at −80 °C and for 3 h at room temperature. The reinjection of plasma and saliva samples after storage for 24 h in the autosampler revealed that the final extract was stable. The CV of the measured concentration of the six spiked individual plasma and saliva samples was below 15% at all investigated concentration levels (QC-L and QC-H).

### 2.8. NMR Metabolomics

Plasma samples were analyzed for their metabolites and lipoproteins by NMR spectroscopy according to Bruker’s standardized and certified in vitro diagnostic research protocol (IVDr; Bruker BioSpin, Billerica, MA, USA). Samples were processed and measured following a strict standard operating procedure, which was previously described by Dona et al. [45]. In brief, thawed aliquots were homogenized with 75 mM sodium phosphate buffer (1/1, *v*/*v*, pH 7.4) by manual shaking and transferred into a 5 mm NMR tube. Prior to analysis, samples were stored at 6 °C in an automated SampleJet™. NMR analysis was performed on a Bruker 600 MHz Avance III HD nuclear magnetic resonance spectrometer with a TXI probe at 37 °C (310 K). Temperature calibration and quantification, as well as water suppression performance, were checked, and settings were optimized before starting the experiments if necessary. Based on the IVDr protocol, a one-dimensional (1D) NOESY experiment (pulse program: noesygppr1d) and a 1D Carr–Purcell–Meiboom–Gill spin-echo experiment (CPMG; pulse program: cpmgpr1d) for the suppression of macromolecules and proteins were measured per sample. The evaluation and quantification of selected parameters were performed according to Bruker quantification in plasma/serum B.I.Quant-PS 2.0.0 and Bruker IVDr Lipoprotein Subclass Analysis (B.I.-LISA). In total, 39 metabolites (+two technical additives) and 112 lipoprotein parameters were quantified automatically (Bruker BioSpin).

### 2.9. Data Analysis and Statistics

The data analysis was conducted using GraphPad version 9.4.0. The normal distribution of the pharmacokinetic data was tested using the D’Agostino–Pearson (omnibus K2) and Anderson–Darling methods. The area under the curve (AUC) from the first to the last sample of the respective intervention phase (AUC_0-tlast_), the maximal achieved concentration (c_max_), and the time at which it appeared (t_max_) were examined to compare the verum and placebo interventions. The data were plotted for each determined substance in plasma and saliva with their mean and SD for each sampling time point.

The individual profiles of spermidine and spermine in plasma can be found in Appendix A. The correlation between the plasma and saliva values of each substance was examined using Spearman’s rank correlation coefficient. Regarding the fasting values, a verum value was compared with the corresponding placebo value, depending on the normal distribution, using a paired t test or a Wilcoxon matched-pairs signed-rank test. A Friedmann test was performed to classify the fasting verum values after multiple administrations of spermidine. The same was performed for the fasting placebo values to compare the verum values with the placebo trend. The differences were considered significant at *p* ≤ 0.05.

Sample 11 of the verum arm was compared with sample 11 of the placebo arm (8 h after first administration), and sample 25 of the verum arm was compared with sample 25 of the placebo arm (the last samples of each intervention phase). The respective samples from the two intervention phases (t = 0 h) were also compared to test the setup and baseline determinations. In addition, a Wilcoxon matched-pairs signed-rank test was performed to detect significant differences between the two intervention arms. All metabolites were plotted together in a forest plot to provide an overview of the changes in sample 11 of the verum arm versus sample 11 of the placebo arm, as well as in sample 25 of the verum arm versus sample 25 of the placebo arm (Figure 4 and Figure 5). The relative error with the placebo sample as a reference value was used. Outlier values were detected using the Grubbs test (ESD method) and removed from the analysis; thus, a total of seven data points were removed. Furthermore, a linear regression was applied to detect possible correlations between the change in AUC_0-tlast_ spermine plasma and metabolome. Therefore, the verum–placebo difference for both AUC_0-tlast_ and metabolic parameters from the last sample of the respective intervention phase was calculated (sample 25).

## 3. Results

### 3.1. Samples

Due to unforeseen events (dizziness and personal appointments), a total of four plasma and two saliva samples could not be taken. Only five (0.833%) of a total of 600 sampling times (50 per subject) were outside the targeted time interval. The largest time deviation was 7 min, while the shortest deviation was 2 min. Thus, more than 99% of the samples were taken within the scheduled time interval. No special relevance was attributed to these five samples, and they were analyzed like the other samples. One saliva sample and one plasma sample were not evaluable due to the analytical failure of the LC–MS/MS system.

### 3.2. Pharmacokinetics in Blood Plasma

Initially, the mean concentrations of spermidine, spermine, and putrescine in the plasma and saliva in the placebo intervention of all 12 subjects combined over all time samples were determined as follows: spermidine (plasma) 12.41 ng/mL (SD 1.869), spermine (plasma) 5.102 ng/mL (SD 0.4646), and putrescine (plasma) 7.403 ng/mL (SD 2.130); spermidine (saliva) 236.5 ng/mL (SD 181.5), spermine (saliva) 201.0 ng/mL (SD 76.02), and putrescine (saliva) 4.343 µg/mL. Subsequently, concentrations of spermidine, spermine, and putrescine in plasma and saliva for both the placebo and verum interventions were compared for AUC_0-tlast_, c_max_, and t_max_. Comparing placebo and verum, no significant differences between the plasma concentrations of spermidine and putrescine were found (Figure 4, Table 1, Appendix A). For spermine, plasma AUC_0-tlast_ was significantly increased in the verum group (*p* = 0.0282).

Interestingly, four of the 12 subjects still had higher AUC_0-tlast_ plasma concentrations for spermine in the placebo intervention compared to the verum intervention (data not presented). Noteworthy, fasting plasma spermine concentrations were higher than the postprandial values in both the placebo and verum arms. The mean values of the fasting samples and the remaining postprandial samples of each subject were averaged to a mean value across the 12 subjects. The mean of the fasted matutinal sample’s placebo intervention for spermine in plasma was 5.720 ng/mL (SD 1.205); for the fasted matutinal samples of the verum intervention, it was 6.502 ng/mL (SD 1.492); for the postprandial samples of the placebo intervention, it was 4.940 ng/mL (SD 1.124); and for the postprandial samples of the verum intervention, it was 4.965 ng/mL (SD 1.324). Investigating whether oral spermidine supplementation increased fasting plasma spermine levels (fasted matutinal concentrations of verum vs. placebo at the respective time point; Wilcoxon signed rank test or paired *t* test depending on normal distribution), no significant difference was found (Table 2); however, a trend of a rising difference was observable over the study duration.

For both study arms, a Friedman test (nonparametric ANOVA) was performed to compare the fasted matutinal spermine plasma levels at 0, 24, 48, 72, and 96 h with each other within the specific study treatment (verum or placebo). Within the placebo treatment, a *p* value of 0.7847 (ns) was found, indicating no difference between matutinal concentrations on the days during the placebo treatment. A Dunn’s multiple comparison test did not find a significant change throughout the days when concentrations at 24, 48, 72, and 96 h were compared with starting spermine concentrations at 0 h during placebo treatment. Within the verum group, a *p* value of 0.0823 (ns) was found, indicating a strong trend of a change in fasted matutinal spermine concentrations during spermidine intake without reaching statistical significance. A Dunn’s multiple comparison test also found no significant change throughout the days when concentrations at 24, 48, 72, and 96 h were compared with starting spermine concentrations at 0 h, even though an increase in matutinal concentrations was observed during verum treatment.

### 3.3. Salivary Pharmacokinetics

An analysis of salivary concentrations found no significant changes in AUC_0-tlast_, c_max,_ and t_max_ (Appendix A, Figure 6, and Table 3) for spermidine, spermine, or putrescine.

The correlations between plasma and saliva concentrations of spermidine, spermine, and putrescine were significantly poor (Spearman r < 0.5, *p* > 0.05; (Figure 7 and Figure 8)). 

### 3.4. NMR Metabolomics

When comparing metabolite concentrations in the blood plasma, no significant differences were observed between the verum and placebo interventions (Appendix A). This also applied in particular to the concentrations of glucose and acetone, both of which are metabolites that have been postulated to change due to the application of calorie restriction mimetics such as spermidine.

As demonstrated above, spermine AUC_0-tlast_ plasma concentrations of the mean values significantly differed when comparing the verum and placebo groups. Since four of the 12 subjects exhibited a spermine AUC_0-tlast_ in the placebo intervention greater than in the verum intervention, a correlation analysis of the metabolome and spermine AUC_0-tlast_ was performed. In the linear regression, the differences between the verum and placebo interventions in AUC_0-tlast_ values of spermine plasma concentrations for each subject were compared with the corresponding differences in metabolite concentrations of the sample 25 (the last value of the corresponding intervention phase). A significant difference was found for LDL-3/4/5, LDL FC, LDL PL, LDL-3/4/5 Chol, LDL-4 FC, LDL-3/4/5 PL, LDL-3/4/5 ApoB, and HDL-1 ApoA2.

## 4. Discussion

### 4.1. Brief Synopsis of Key Findings

To the best of our knowledge, this is the first randomized placebo-controlled pharmacokinetic study with spermidine in humans. Previous data on the pharmacokinetics of spermidine are either based on animal studies only or on clinical trials that do not meet the standards of pharmacokinetic studies. Most strikingly, we found that oral intake of spermidine at 15 mg/d for 5 days increased spermine levels in the plasma but not levels of spermidine or putrescine. Similarly, a nonsignificant trend indicated that spermidine supplementation increased fasted matutinal spermine plasma concentrations but not respective spermidine and putrescine concentrations.

### 4.2. In Vivo and Ex Vivo Data

Available pharmacokinetic data for spermidine are mainly limited to in vivo and animal models and suggest contradictory results. In a rodent ex vivo model, the portal vein recovery of ^13^C-labeled spermidine after jejunal instillation was 61–76% within 20 min [19]. In rats, intragastrically administered ^14^C-labeled polyamines were found to be rapidly absorbed in the small intestine, passed into systemic circulation, and distributed to different tissues [9]. In a different setting with rats, gavage of a liquid spermidine/spermine mixture followed by whole blood sampling from both the portal vein and vena cava inferior led to significantly increased concentrations of both spermidine and spermine in whole blood samples, peaking at 1 h after oral administration [46].

In mouse models, prolonged administration of polyamine-rich chow was found to (partly sex-dependently) increase both spermidine and spermine blood levels [17,47]. In contrast, Soda et al. reported that polyamine-rich chow in mice significantly increased spermine blood concentrations but not spermidine concentrations, which is in line with our results [48].

Interestingly, early in vivo experiments in rats with dietary polyamines from the 1990s revealed differential effects of presystemic eliminations. After intragastrical administration, only 11–15% of putrescine was recovered in intestinal tissue, but 87–96% of spermidine and 79–82% of spermine were recovered [49]. Based on these data, it has been suggested that most dietary putrescine is degraded by diamine oxidase and polyamine oxidase, which are both abundant in intestinal tissue. However, dietary spermidine and spermine are almost quantitatively absorbed without catalytic breakdown. Further presystemic processes after absorption remain unclear.

### 4.3. Pharmacokinetic Data from Clinical Trials

Previous analyses of spermidine pharmacokinetics in humans are scarce; published data are not based on randomized placebo-controlled interventions but rather on noninterventional data or food-based interventions. In 2018, the results of a randomized, placebo-controlled, double-blind phase II trial with supplementation of a polyamine-rich plant extract were reported [47]. For 3 months, 30 participants received a supplement providing a daily dose of spermidine (1.2 mg), spermine (0.6 mg), and putrescine (0.2 mg). Blood polyamine concentrations did not differ between placebo controls and spermidine-supplemented individuals. This result can easily be explained by the underdosing of the supplement.

More recently, confirming data from rodent experiments [48], in a nonrandomized intervention trial, Soda et al. investigated the effect of a polyamine-rich food (natto) on spermidine and spermine blood concentrations of young male subjects [50]. In this trial, natto consumption was found to increase the mean spermine levels in whole blood, while blood spermidine levels remained unchanged despite increased dietary intake. Our results are in line with these findings, despite being based on a completely different methodology. Thus, taking together respective data from rodents [48], human food-based data [50], and our results presented here, one can postulate that spermidine, after oral intake, rather acts as a prodrug that is converted to spermine by spermine synthase. Consequently, previously described health-related effects after spermidine supplementation are likely to be mediated not by spermidine itself but by increased spermine concentrations. Notwithstanding, it might be possible that systemically distributed spermine is subsequently reconverted to spermidine by spermine oxidase (SMOX) or spermine synthase (SMS) in peripheral tissues. 

### 4.4. Salivary Concentrations of Polyamines

To our knowledge, no data on the salivary pharmacokinetics of spermidine, spermine, and putrescine after oral supplementation of spermidine have been published before. In our study, neither the salivary AUC_0-tlast_, c_max_, nor t_max_ differed significantly between the verum and placebo treatments for the three polyamines investigated. Furthermore, no significant correlation between plasma and salivary concentrations of spermidine, spermine, or putrescine was detected. 

Polyamines, including spermidine and putrescine, reportedly have diverse functions in the bacterial physiology of the oral microbiota [51]. However, the role of the oral microbiota in systemic polyamine metabolism has yet to be elucidated [52]. To date, it remains speculative whether a differential oral microbiota composition may contribute to the differential effects of oral spermidine intake on salivary polyamine concentrations.

As described earlier, a reduction of the viral replication of SARS-CoV-2 by 85% was demonstrated in vitro at a concentration of 333 µM spermidine, encouraging discussions about the preventive or even therapeutic potential of spermidine in the context of COVID-19 [38]. Our results, however, do not support this approach, as even the highest individual salivary spermidine concentration of 3191 ng/mL, corresponding to 22.0 µM, is smaller by more than a factor of 10 than the aforementioned effective in vitro concentration. Thus, oral spermidine supplementation does not seem to be an effective approach for reducing the risk of an infection with SARS-CoV-2. Higher spermidine concentrations in the pharynx could possibly be achieved using alternative application methods, such as a spray, mouth rinsing and gargling, or inhalation.

Furthermore, limited data are available on salivary concentrations of spermidine, spermine, and putrescine without oral spermidine supplementation. Cooke et al. investigated the time profiles of putrescine, cadaverine, indole, and skatole upon waking and during the day [53]. In their study, the highest concentrations of putrescine (mean 33.0 µg/mL) were measured immediately upon waking. In the present study, the highest concentrations of polyamines were also observed in fasted matutinal salivary probes. After breakfast and brushing, the salivary putrescine concentration fell to 7.0 µg/mL, while during the day an increase was observed. These high matutinal salivary polyamine concentrations are probably caused by bacterial putrefaction [53]. Proteins and peptides present in the oral cavity are broken down into amino acids, which are further decomposed by various enzymes. For example, ornithine can be converted to putrescine by ornithine decarboxylase. High amounts of putrescine may be processed to spermidine by spermidine synthase and further to spermine by spermine synthase [54]. During sleep, spermidine, spermine, and putrescine could accumulate in the oral cavity, and, after waking up, eating meals and oral cleaning would lead to a decrease in salivary polyamine concentrations.

Li et al. measured the salivary concentration of various polyamines and amino acids in five healthy volunteers before and after they brushed their teeth [55]. Before brushing, the concentration of spermidine in saliva was 3.40–10.84 µmol/L (493.9–1575 ng/mL), whereas after brushing it was in a range of 1.61–6.41 µmol/L (233.9–931.1 ng/mL). Salivary putrescine concentrations were in the range of 0.54–133.36 µmol/L (47.6–11,756 ng/mL) before and 0.26–3.64 µmol/L (22.9–320.9 ng/mL) after brushing teeth. Several research groups have investigated the salivary concentrations of polyamines as a possible biomarker for various types of cancer [56,57,58,59]. Usually, cancer patients have been demonstrated to have elevated salivary polyamine concentrations. However, even in healthy subjects, the variability of salivary polyamine concentrations across different studies is apparently rather high. Venza et al. investigated the salivary polyamine concentrations in 50 healthy subjects and divided the subjects into five different groups according to age and/or gender [60]. No association between age or gender and salivary polyamine concentrations was found. The determined average salivary concentrations of the groups of 107–110 ng/mL for spermidine and 118–121 ng/mL for spermine were in the same order of magnitude as those in our study. However, Venza et al. found average salivary putrescine concentrations of 24–29 ng/mL, much lower concentrations than our study found. In another study, the mean salivary concentrations of spermidine and spermine of 14 healthy subjects were below 10 ng/mL, while the mean salivary putrescine concentrations were approximately 130 ng/mL with noticeable interindividual variability [56]. Takayama et al. found the mean salivary concentrations of 61 healthy subjects to be approximately 0.55 nmol/mL (79.9 ng/mL) for spermidine, 0.055 nmol/mL (11.1 ng/mL) for spermine, and 15.5 nmol/mL (1366 ng/mL) for putrescine [57]. Dame et al. reported average salivary concentrations of spermidine of 3.21 µmol/L (466.3 ng/mL), of spermine of 0.63 µmol/L (127.5 ng/mL), and of putrescine of 136.76 µmol/L (12,055 ng/mL) for 16 healthy volunteers [61]. Overall, both the inter- and intraindividual variability of salivary polyamine concentrations was quite high in our study, but pharmacokinetics related to supplementation were not evaluable.

### 4.5. Relevance of Presystemic Conversion of Spermidine

Although it has been speculated that intestinal polyamine uptake might be similar in rodents and humans, our data indicate that—different from rodent data—dietary spermidine is not quantitatively recovered in the systemic circulation but is subject to extensive presystemic metabolism. These findings are supported by previous studies. For instance, Soda et al. demonstrated millimolar concentrations of dietary polyamines in the duodenum and jejunum in humans after high-dose oral intake yielded maximal plasma concentrations of only 10–20 μM [17]. Most likely, after enterocytic uptake, spermidine is almost completely converted to spermine, either in the enterocytes or by hepatic first-pass metabolism. The mechanisms of spermidine degradation to spermine were investigated and described in detail elsewhere [8]. Predominantly, spermidine is converted to spermine by spermine synthase. Alternatively, spermidine is converted to acetyl-spermidine by spermidine/spermine-N(1)-acetyltransferase, which is then degraded to putrescine by N(1)-acetyl-spermine/spermidine oxidase or microsomal acetylpolyamine oxidase. Thus, putrescine is both a biological precursor of spermidine and its metabolite. Nonetheless, we observed no changes in putrescine concentrations in saliva or plasma.

Both the enterocytic and hepatic conversion of spermidine to spermine might explain the increased spermine concentration in blood plasma after oral spermidine intake, while spermidine concentrations remained unchanged. Alternative explanations could be a complete intraluminal degradation, a lack of intestinal absorption, or a rapid distribution to tissues and organs with peripheral uptake rates being similar to intestinal absorption, which would leave blood concentrations at a constant level. Since none of these mechanisms can explain the observed increase in spermine levels over time, our data strongly suggest that in humans, dietary spermidine is at least partly presystemically converted into spermine, which then occurs in the systemic circulation. Since no spermine pharmacokinetics were evaluable, these processes are probably concealed by other metabolic or distributing processes, as discussed above.

### 4.6. Role of the Intestinal Microbiota

Evidently, the pharmacokinetic relationship between oral spermidine intake and spermidine/spermine plasma concentrations is not simply additive, and the underlying mechanisms remain rather speculative. Obviously, dietary spermidine is subject to extensive presystemic metabolism. In this context, it has been suggested that the intestinal microbiota is involved in the presystemic metabolism of dietary polyamines. Matsumoto et al. reported that the administration of probiotic bifidobacteria increased spermine—but not spermidine—concentrations in the feces of animals and humans [62,63]. In fact, polyamine production by intestinal bacteria is a main source of exogenous polyamines, and dietary arginine (a precursor of polyamine biosynthesis) was demonstrated to increase both colonic putrescine concentration and blood levels of spermidine and spermine [63]. However, as dietary polyamines have been reported to be quantitatively absorbed in the duodenum and the proximal jejunum, the involvement of the colonic microbiota in the presystemic metabolism of dietary spermidine appears to be rather unlikely.

### 4.7. NMR Metabolomics

As demonstrated above, a positive correlation of plasma spermine concentrations with several cholesterol fractions was found. A causal effect due to spermidine supplementation cannot simply be postulated, since no significant differences were observed between the verum and placebo interventions (Appendix A). While these results could be the basis for future research, they should not be discussed further in the context of the present pharmacokinetic study.

### 4.8. Clinical and Research Implications

Since spermidine supplementation has been postulated as a possible means of therapeutically targeting several diseases [3,7], translational approaches with spermidine are a hot topic. Therefore, our results have considerable significance for future clinical research.

First, our data demonstrate that a daily dose of 15 mg of spermidine significantly increases blood concentrations of spermine. In previous trials, daily spermidine doses of 0.9–1.2 mg have been used without any change in blood concentrations of polyamines or any clinical efficacy [47,64]. Taking our data into account, those results may have been a consequence of simple underdosing. It is rather unlikely that spermidine supplements with daily doses below 15 mg exert any effect, since even the effect observed in this study was rather small. Thus, in future trials, daily doses of at least 15 mg should be used. While it is not impossible that long-term supplementation of low-dose spermidine (<15 mg/d) exerts any effect through epigenetic modulations or metabolic cascades, such effects remain highly speculative. 

Second, it can be postulated that the in vitro and clinical effects of spermidine are not solely due to spermidine itself but may also be attributed to its metabolite, spermine, which is formed in the body prior to or in peripheral tissues [64]. The concept of spermidine mainly representing the prodrug of biologically active spermine is further supported by the pathophysiology of Snyder–Robinson syndrome (SRS) [65]: SRS is a very rare X-linked recessive condition caused by a mutation in the spermine synthase gene located at Xp.21.3-p22.12. This mutation causes incorrect splicing that results in an inactive truncated protein, leading to the loss of most spermine synthase activity (>90%). Affected patients exhibit numerous adverse health effects, including intellectual disability, muscle hypoplasia, facial dysmorphism, renal abnormalities, and seizures [66,67]. Most interestingly, spermine is significantly reduced in SRS, whereas spermidine is substantially increased.

Third, the overall view of preclinical and clinical data, including the results presented here, suggests that the mechanisms of intestinal absorption as well as the mechanisms of presystemic spermidine metabolism are different in rodents and humans, thereby directly affecting the biological effects. Therefore, special caution is required when transferring experimental animal data with spermidine to humans. On the other hand, spermine might represent the predominant form of transport of spermidine in the blood, with dietary spermidine being absorbed into enterocytes and intracellularly converted to spermine, which is then extruded into the systemic circulation through the basolateral membrane, followed by reconversion to spermidine in tissues and organs. It is well known that the conversion of spermidine to spermine is not a one-way street but rather possible in both directions [8]. Nonetheless, these processes were not visible in spermine pharmacokinetics after spermidine supplementation.

Fourth, oral spermidine intake of 15 mg/d does not affect salivary polyamine concentrations. Preclinical data implying an oropharyngeal effect of spermidine may not be translated into recommendations to supplement spermidine, as effective doses have yet to be determined. Systemic doses of 15 mg/d do not seem to be appropriate for yielding pharmacological concentrations in the saliva.

Fifth, our findings challenge the concept of epidemiological spermidine studies based on statistical correlations between the spermidine content of food and biological effects without considering the variability of presystemic spermidine metabolism or dietary spermine intake.

### 4.9. Limitations of the Study

There are a number of limitations that should be considered in interpreting the current findings. Since this study was designed as an exploratory pharmacokinetic trial, case number calculation was not possible due to a lack of data. Accordingly, a sample size of 12 was considered appropriate as it was established in phase 1 clinical trials. Therefore, it cannot be excluded that the study was not sufficiently powered to detect minor concentration changes. However, the robustness of the data was improved by the cross-over design with placebo control, which can be considered the baseline. 

Validated analytic results of spermidine, spermine, and putrescine can be estimated as robust but were performed in plasma after protein precipitation. It is unclear whether the substances could be transported intracellularly in blood cells or bound to plasma proteins in relevant amounts, which would have led to an underestimation of the complete amount present in whole blood. It is also possible that there might be some significant effects of spermidine supplementation if taken by special subgroups, in higher doses, or simultaneously with certain foods. Moreover, the intervention phases were rather short (5-day interventions), and it cannot be excluded that a long-term intake of spermidine would show deviating results. The latter seems unlikely, as Soda et al. obtained very similar results on spermine concentrations with similar amounts of supplemented spermidine, even after long-term administration [50].

## 5. Conclusions

The results presented here for the first time provide a pharmacokinetic basis for future translational research on spermidine. Oral spermidine intake of 15 mg/d for 5 days significantly increased spermine levels in the plasma but did not affect spermidine or putrescine levels. Our data strongly suggest that dietary spermidine is presystemically converted into spermine, which then occurs in the systemic circulation. Consequently, we postulate that the in vitro and clinical effects of spermidine are (at least in part) not attributable to spermidine itself but rather to its metabolite, spermine. It is rather unlikely that spermidine supplements with doses <15 mg/d exert any effect. Moreover, even spermidine doses of 15 mg/d do not affect salivary polyamine concentrations; in particular, pharmacological spermidine concentrations in the saliva that, due to preclinical data, might be effective at inhibiting oropharyngeal SARS-CoV-2 replication are not remotely achieved.

Finally, epidemiological studies that have correlated dietary spermidine intake with biological effects are challenged by our results, as such studies have not considered the variability of presystemic spermidine metabolism and dietary spermine intake.

## Figures and Tables

**Figure 1 nutrients-15-01852-f001:**
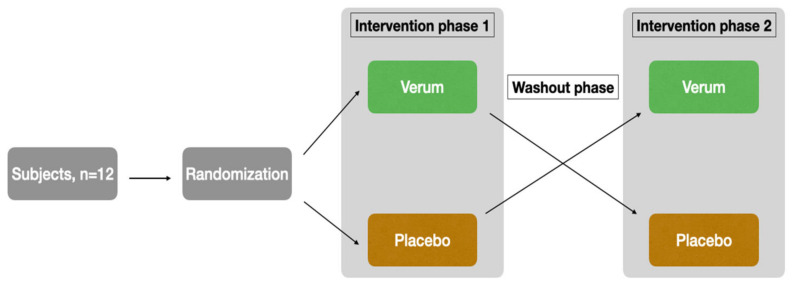
Study design.

**Figure 2 nutrients-15-01852-f002:**
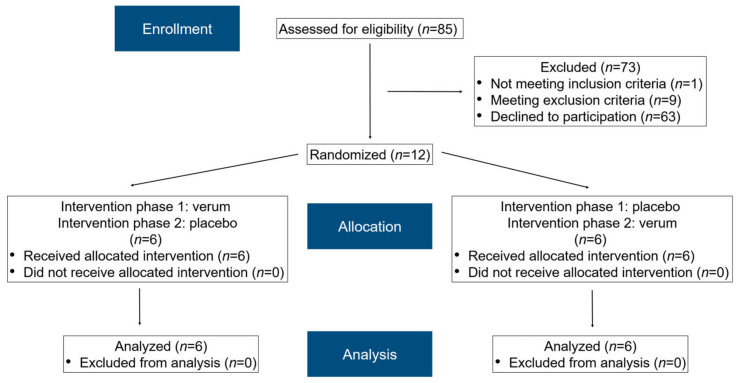
A modified flow chart of the randomization phases according to CONSORT 2010 [43].

**Figure 3 nutrients-15-01852-f003:**
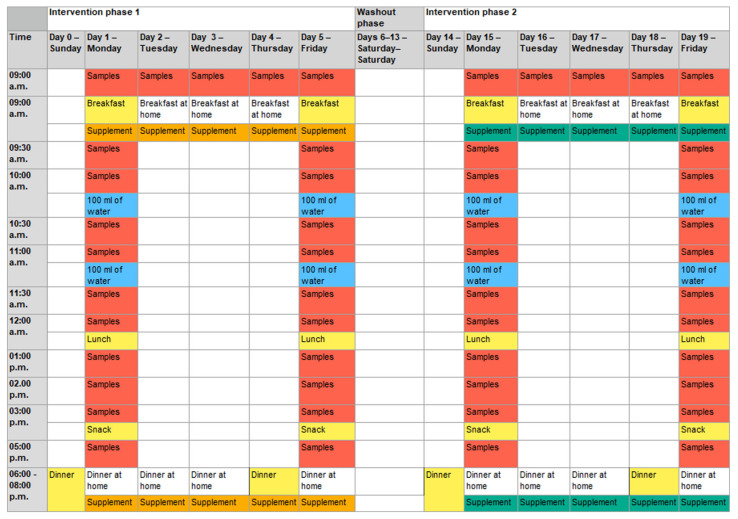
Detailed sequences of the two study phases. Depending on the randomization, the placebo was used first in intervention phase 1, followed by verum supplementation in phase 2, or vice versa. Five capsules: intake of five capsules of placebo or verum with the first bite of the meal; samples: blood and saliva samples were taken; breakfast: standard breakfast; breakfast at home: self-selected, non-standardized breakfast at home; lunch: standard lunch; snack: standard snack; dinner: standard dinner; dinner at home: self-selected, non-standardized dinner at home; 100 mL of water: intake of 100 mL of water immediately after taking the sample at the respective time. After lunch, water intake was provided ad libitum. Colored days/times mark study-related activities at the study center. Days/times with study-related activities at home or without any study-related activities are not colored.

**Figure 4 nutrients-15-01852-f004:**
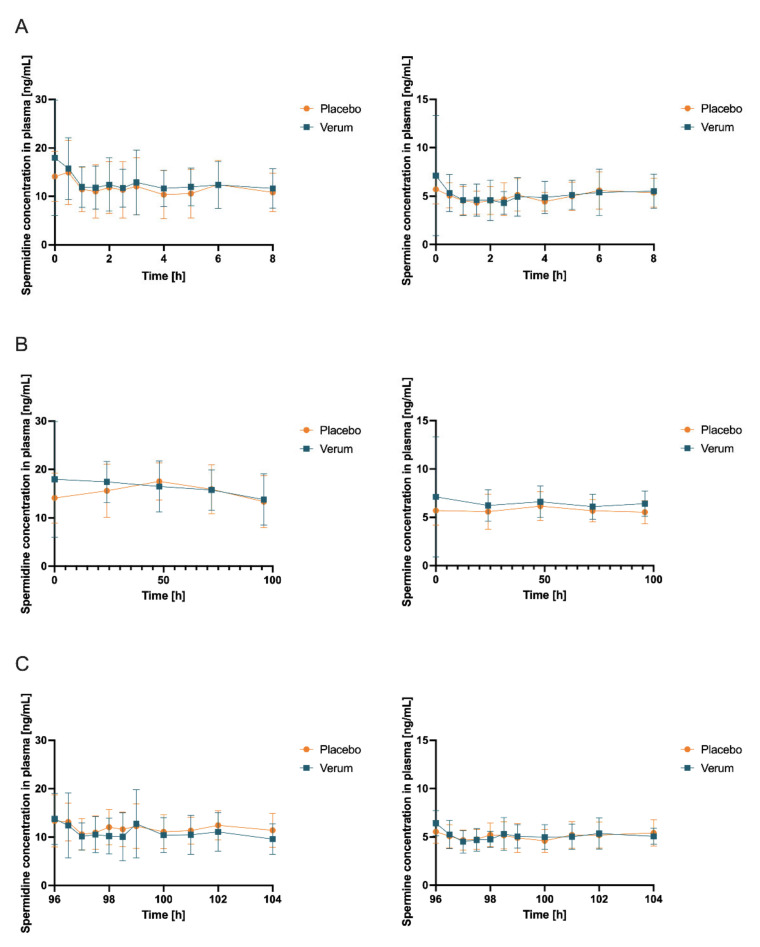
Concentrations (mean ± SD, *n* = 12) of spermidine (left) and spermine (right) in plasma with time point t = 0 h corresponding to first intake of respective allocated treatment and respective baseline concentration obtained immediately before intake (**A**) at first measurement day of verum or placebo treatment, respectively (which can be day 1 or day 15 due to randomization), 0–8 h after first administration of verum or placebo; (**B**) first samples of days 1, 5, 15, and 19 and days 2, 3, 4, 16, 17, and 18 (all fasted matutinal samples of each intervention phase); (**C**) day 5 and day 19 (last measurement day of each intervention phase which can be verum or placebo due to randomization), 96–104 h after multiple administration of verum or placebo.

**Figure 5 nutrients-15-01852-f005:**
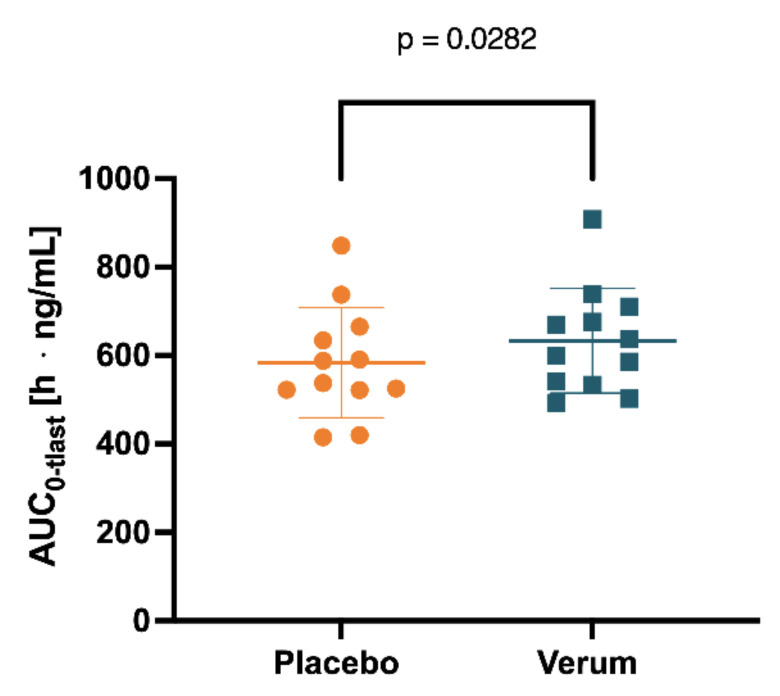
Spermine concentration in plasma measured for 0–104 h was determined by AUC_0-tlast_ after placebo and verum interventions.

**Figure 6 nutrients-15-01852-f006:**
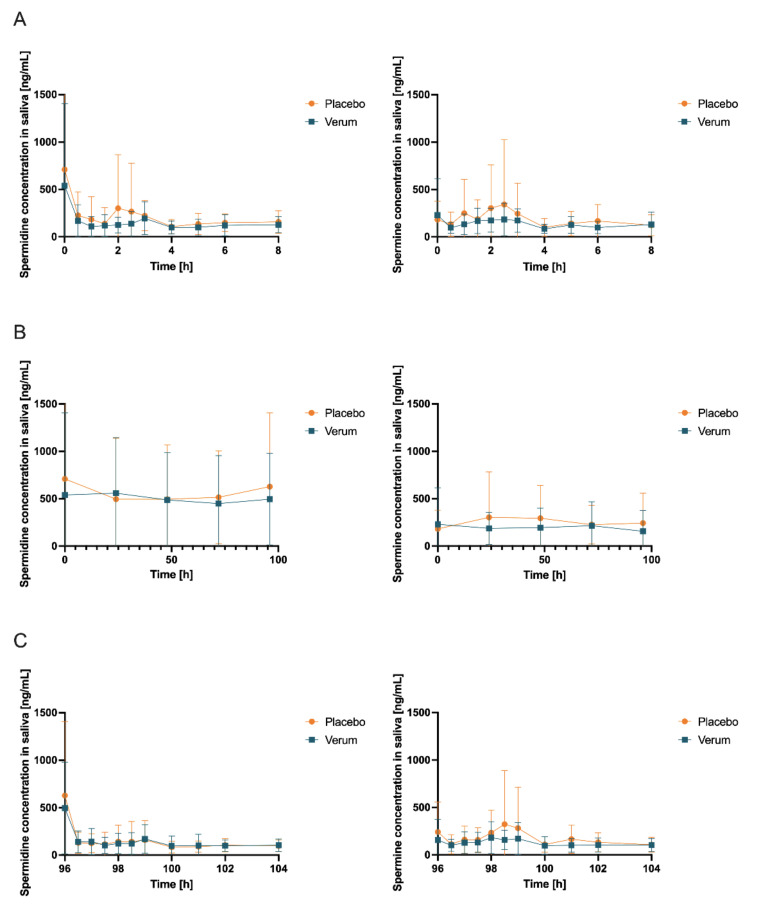
Salivary concentrations (mean ± SD, *n* = 12) of spermidine (left) and spermine (right); (**A**) day 1 and day 15 (first measurement day of each intervention phase), 0–8 h after first administration of verum or placebo; (**B**) first sample days 1, 5, 15, and 19 and days 2, 3, 4, 16, 17, and 18 (all fasted matutinal samples of each intervention phase); (**C**) day 5 and day 19 (last measurement day of each intervention phase), 96–104 h after multiple administration of verum or placebo.

**Figure 7 nutrients-15-01852-f007:**
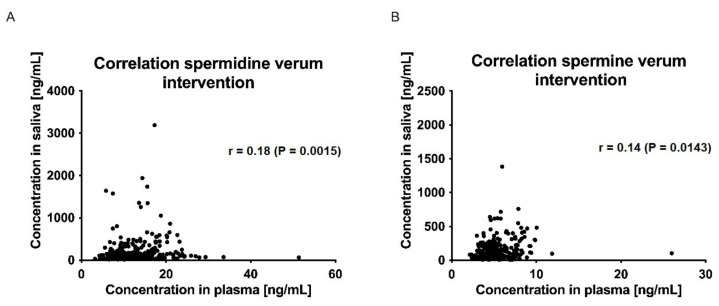
Correlation of saliva and plasma concentrations of spermidine (**A**) and spermine (**B**) for the verum intervention over all time points. Data were analyzed using Spearman’s rank correlation coefficient.

**Figure 8 nutrients-15-01852-f008:**
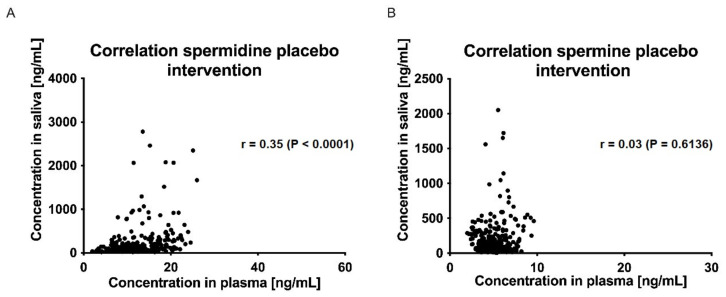
Correlation of saliva and plasma concentrations of spermidine (**A**) and spermine (**B**) for the placebo intervention over all time points; data were analyzed using Spearman’s rank correlation coefficient.

**Table 1 nutrients-15-01852-t001:** The plasma spermidine and spermine mean and SD of AUC_0-tlast,_ c_max_, t_max_, and *p* value for placebo versus verum.

Parameter	Substance	Placebo	Verum	*p* Value
AUC_0-tlast_ mean (SD) h · ng/mL	Spermidine	1533 (406.2)	1567 (412.6)	0.7819
	Spermine	583.8 (124.8)	633.0 (118.5)	0.0282
c_max_ mean (SD) ng/mL	Spermidine	19.65 (3.985)	22.70 (10.67)	0.7754
	Spermine	7.19 (1.19)	9.45 (5.47)	0.1294
t_max_ mean (SD) h	Spermidine	53.13 (38.97)	32.29 (31.78)	0.2317
	Spermine	57.83 (41.29)	57.88 (39.61)	0.9983

**Table 2 nutrients-15-01852-t002:** *p* values from the Wilcoxon signed rank test or paired *t* test derived from the comparison of the respective fasted matutinal placebo measurements with the associated fasted matutinal verum measurements of spermine in plasma.

Comparative Sample Time Points	*p* Value: Placebo vs. Verum Measurement
0 h	0.9097
24 h	0.3298
48 h	0.3804
72 h	0.1392
96 h	0.0753

**Table 3 nutrients-15-01852-t003:** Salivary spermidine and spermine AUC_0-tlast_, c_max_, t_max_, and *p* value for placebo versus verum. Data are presented as the mean (SD).

Parameter	Substance	Placebo	Verum	*p* Value
AUC_0-tlast_ mean (SD) h · ng/mL	Spermidine	46,470 (48,717)	42,707 (41,636)	0.2661
	Spermine	25,177 (28,371)	18,546 (17,021)	0.1514
c_max_ mean (SD) ng/mL	Spermidine	927.2 (951.9)	751.1 (884.8)	0.1763
	Spermine	523.7 (678.0)	389.5 (386.3)	0.5186
t_max_ mean (SD) h	Spermidine	40.92 (37.77)	52.00 (38.06)	0.5072
	Spermine	42.58 (44.11)	47.75 (44.32)	0.6719

## Data Availability

Not applicable.

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
