# Peer review of "High-Dose Spermidine Supplementation Does Not Increase Spermidine Levels in Blood Plasma and Saliva of Healthy Adults: A Randomized Placebo-Controlled Pharmacokinetic and Metabolomic Study"

_nutrients, 2023, doi:10.3390/nu15081852_

Round 1

Reviewer 1 Report

In the present study, Senekowitsch et al. performed a randomized, placebo-controlled, triple-blinded, two-armed crossover trial to evaluate the level of polyamines in both serum and saliva after systemic supplementation of spermidine. Overall, it is a well-designed study and the pharmacokinetics of orally administered spermidine can be instructive for further studies.

Here are some comments and suggestions: 

1.     The introduction of this essay should be more concise.

2.     Please introduce the blind method and allocation concealment you used in the study in detail.    

3.     Add the baseline table of included patients, especially their clinical characters. For instance, serum spermidine is supposed to be positively associated with obesity (PMID: 35807793; DOI: 10.3390/nu14132613), so, please describe the demographic characteristics of these participants.

4.     Please discuss the limitation of the study.

5.     Besides the potential application of spermidine in the central nerve system, the circulation system, and the digestive system, its role in ocular diseases should also be discussed in the introduction (for example PMID: 35890394, DOI: 10.3390/pharmaceutics14071500).

Author Response

Dear Reviewer,

Thank you very much for the careful review of our manuscript and the valuable advice for improvement. Please find the details of our changes below.

Revisions:

C1: The introduction of this essay should be more concise.

A1: Thank you very much for this advice. We revised the introduction in the suggested way and shortened it by 121 words.

C2: Please introduce the blind method and allocation concealment you used in the study in detail.    

A1: Chapter 2.1 was amended by the following paragraph:

With the command =random number(), a random number between 0 and 1 was generated for each subject. Subsequently, a comparison was made in blocks of two to determine which subject had the higher random number. Those with the higher random number started with the verum arm, while subjects with the lower random number started with the placebo arm. This procedure ensured an equal distribution (balance) of the two possible orders V-P or P-V in 12 subjects. Thus, 6 subjects each start with verum or placebo. The capsules taken (verum or placebo) were provided by staff in Greifswald on the basis of the generated sequence for week 1 and week 2 specifically for the subject-specific random number. Externally, verum and placebo are indistinguishable due color matched filling powder and capsules of the same size and color. For more information about products and capsules see chapter 2.3. The containers were assigned on the basis of pseudonym and subject number as well as treatment week. Thus, at the study site in Lübeck, the performing staff as well as the subjects were blinded to the assignment and only received the respective prepared capsule container for week 1 or week 2. Until the analyses were completed, the samples and results were only assigned to week 1 or week 2 of the subjects. Only after completion of the measurements, the unblinding was carried out using the randomization list.

C3: Add the baseline table of included patients, especially their clinical characters. For instance, serum spermidine is supposed to be positively associated with obesity (PMID: 35807793; DOI: 10.3390/nu14132613), so, please describe the demographic characteristics of these participants.

A3: Clinical parameters like body weight, body mass index (BMI), or waist hip ratio (WHR) have not been determined because of the crossover design of this pharmacokinetic study.

C4: Please discuss the limitation of the study.

A4: Thank very much for this important point. A separate paragraph on the limitations of the study has been added:

4.9 Limitations of the study

There are a number of limitations that should be considered in interpreting the current findings. Since this study was designed as an exploratory pharmacokinetic trial, case number calculation was not possible due to a lack of data. Accordingly, a sample size of 12 was considered appropriate as it is established in phase 1 clinical trials. Therefore, it cannot be excluded that the study was not sufficiently powered to detect minor concentration changes. However, the robustness of the data was improved by the cross-over design with placebo control which can be considered as baseline. Validated analytic of spermidine, spermine and putrescin can be estimated as robust, but was performed in plasma after protein precipitation. It is unclear whether the sub-stances could be transported intracellularly in blood cells or bound to plasma proteins in relevant amounts, which would have led to an underestimation of complete amount present in whole blood.

It is also possible that there might be some significant effects of spermidine supplementation, if taken by special subgroups, in higher doses, or simultaneously with certain foods. Moreover, the intervention phases were rather short (5-day interventions), and it cannot be excluded that a long-term intake of spermidine would show deviating results. The latter seems unlikely as Soda et al. obtained very similar results on spermine concentrations with similar amounts of supplemented spermidine, even after long-term administration.

C5: Besides the potential application of spermidine in the central nerve system, the circulation system, and the digestive system, its role in ocular diseases should also be discussed in the introduction (for example PMID: 35890394, DOI: 10.3390/pharmaceutics14071500).

A5: The reference to the possible importance of spermidine in ophthalmology was added to the introduction.

Reviewer 2 Report

Senenkowitsch et al. examine the impact of spermidine supplementation on spermidine levels in blood plasma and saliva of healthy adults.  The study is organized as a randomized, placebo-controlled, triple-blinded two-armed crossover trial.  Blood and saliva levels of spermidine, spermine, and putrescine were quantified by LC-MS/MS.  In addition, the plasma metabolome was evaluated using NMR metabolomics. There were 12 total subjects.  Subjects were of normal BMI, healthy, and young adults. I have provided some comments below for potential improvement listing in order roughly from major-to-minor concerns.

11.       Figure 3 (Sequence)- It isn’t clear whether the water (100 ml) is taken every day just after taking the supplement (as described in the legend) or after two different samples are collected.  Please revise figure to clarify.  Why are some days colored in and other left blank?  What is meant by “breakfast at home” etc. in the legend.  How does this correspond to anything in the Figure?

22.      Abstract- “which than occurs in the systemic circulation”, should be “which then…”

33.       Methods (page 6)- What is “still” water?  “distilled”?

44.       Was there any attempt to adjust the standard meals based on estimated basal metabolic rate?

55.       Figure 4. Needs some clarification- It is unclear which is Day 1 and which is Day 15.  Also what does the hours on the X-axis correspond to?  Is this hours from the initial dose?  Does the “0” hour point reflect sampling after taking the Verum (or placebo) or prior (baseline)? 

66.       Figure 7- the correlation figure could be improved with color-coding of time points after starting treatments and correlation coefficients included in the inset of the figure.

77.       I think I would change the title to “Oral intake of spermidine increases spermine but not spermidine or putrescine levels in blood and saliva” to emphasize the change that you did find.

88.       One figure should be provided on the metabolomics.  Setting the criteria for “significance” lower may allow you to find some trends in metabolites that could be informative.  If space is an issue, you could do away with Figure 1 and move it to the Supplement.

Author Response

Dear Reviewer,

Thank you very much for the careful review of our manuscript and the valuable advice for improvement. Please find the details of our changes below.

Revisions:

C1: Figure 3 (Sequence)- It isn’t clear whether the water (100 ml) is taken every day just after taking the supplement (as described in the legend) or after two different samples are collected.  Please revise figure to clarify.  Why are some days colored in and other left blank?  What is meant by “breakfast at home” etc. in the legend.  How does this correspond to anything in the Figure?

A1: Thank you very much for these valuable advices.

Regarding the water intake, the legend of Figure 3 has been clarified. Furthermore, an explanation for why some days are colored and others are left blank has been added to the legend.

The meaning of “breakfast at home” etc. has been clarified in the legend, too. It is defined as “self-selected, non-standardized breakfast at home”. It corresponds to “breakfast at home” in the figure.

C2: Abstract- “which than occurs in the systemic circulation”, should be “which then…”

A2: Has been corrected.

C3: Methods (page 6)- What is “still” water?  “distilled”?

A3: Has been replaced by “non-carbonated water”.

C4: Was there any attempt to adjust the standard meals based on estimated basal metabolic rate?

A4: No, the standard meals have not been adjusted to the individual estimated basal metabolic rate, because due to the cross-over design of the study this would have been without impact on the pharmacokinetic data.

C5: Figure 4. Needs some clarification- It is unclear which is Day 1 and which is Day 15.  Also what does the hours on the X-axis correspond to?  Is this hours from the initial dose?  Does the “0” hour point reflect sampling after taking the Verum (or placebo) or prior (baseline)? 

A5: We do agree that this Figure is hard to understand due to complexicity of study design and measurement time points. Thus, we have modified the caption of Figure 4.

Concentrations (mean ± SD, n = 12) of spermidine (left) and spermine (right) in plasma (A) at first measurement day of verum or placebo treatment respectively (which can be day 1 or day 15 due to randomization), 0–8 h after first administration of verum or placebo; (B) first samples of days 1, 5, 15, and 19 and days 2, 3, 4, 16, 17, and 18 (all fasted matutinal samples of each intervention phase); (C) day 5 and day 19 (last measurement day of each intervention phase which can be varum or placebo due to randomization), 96–104 h after multiple administration of verum or placebo with time point t = 0 h corresponding to first intake of respective allocated treatment and respective baseline concentration obtained immediately before intake.

C6: Figure 7- the correlation figure could be improved with color-coding of time points after starting treatments and correlation coefficients included in the inset of the figure.

A6: Only time points after start of the verum treatment were included to the analysis and the respective Figure 7, thus color-coding would not add additional information to our understanding.

We agree that implementation of the specific correlation coefficients might be helpful, so that we have included them.

Same applies for Figure 8.

C7: I think I would change the title to “Oral intake of spermidine increases spermine but not spermidine or putrescine levels in blood and saliva” to emphasize the change that you did find.

A7: Thank you for this plausible suggestion, which is quite understandable. Nevertheless, we would prefer to leave the title as submitted (“High-Dose Spermidine Supplementation Does Not Increase Spermidine Levels in Blood Plasma and Saliva”) in order to emphasise that the expected effect of spermidine supplementation (an increase in spermidine concentrations) does NOT occur.

C8: One figure should be provided on the metabolomics.  Setting the criteria for “significance” lower may allow you to find some trends in metabolites that could be informative.  If space is an issue, you could do away with Figure 1 and move it to the Supplement.

A8: Thank you very much for this very valuable advice and your interest in metabolomics. However, we would like to keep the metabolomics data in this manuscript rather short, because many more details must be considered if putting these data to a more prominent place. Therefore, we would like to address and discuss the metabolomics in depth in another publication.

Round 2

Reviewer 2 Report

The revisions are all appropriate.